# The Feasibility of Host Transcriptome Profiling as a Diagnostic Tool for Microbial Etiology in Childhood Cancer Patients with Febrile Neutropenia

**DOI:** 10.3390/ijms21155305

**Published:** 2020-07-26

**Authors:** Martina Wahlund, Indranil Sinha, Kristina Broliden, Shanie Saghafian-Hedengren, Anna Nilsson, Anna Berggren

**Affiliations:** 1Department of Medicine Solna, Infectious Disease Unit, Center for Molecular Medicine, Karolinska University Hospital, Karolinska Institutet, 171 76 Stockholm, Sweden; martina.wahlund@ki.se (M.W.); kristina.broliden@ki.se (K.B.); 2Clinical Microbiology, Karolinska University Hospital, 171 76 Stockholm, Sweden; 3Childhood Cancer Research Unit, Department of Women’s and Children’s Health, Karolinska Institutet, 171 76 Stockholm, Sweden; Indranil.Sinha.2@ki.se (I.S.); shanie.hedengren@ki.se (S.S.-H.); Anna.Nilsson.1@ki.se (A.N.); 4Astrid Lindgren Children’s Hospital, Karolinska University Hospital, 171 76 Stockholm, Sweden

**Keywords:** transcriptome, infection, children, neutropenia

## Abstract

Infection is a common and serious complication of cancer treatment in children that often presents as febrile neutropenia (FN). Gene-expression profiling techniques can reveal transcriptional signatures that discriminate between viral, bacterial and asymptomatic infections in otherwise healthy children. Here, we examined whether gene-expression profiling was feasible in children with FN who were undergoing cancer treatment. The blood transcriptome of the children (*n* = 63) was investigated at time of FN diagnosed as viral, bacterial, co-infection or unknown etiology, respectively, and compared to control samples derived from 12 of the patients following the FN episode. RNA sequencing was successful in 43 (68%) of the FN episodes. Only two genes were significantly differentially expressed in the bacterial versus the control group. Significantly up-regulated genes in patients with the other three etiologies versus the control group were enriched with cellular processes related to proliferation and cellular stress response, with no clear enrichment with innate responses to pathogens. Among the significantly down-regulated genes, a few clustered into pathways connected to responses to infection. In the present study of children during cancer treatment, the blood transcriptome was not suitable for determining the etiology of FN because of too few circulating immune cells for reliable gene expression analysis.

## 1. Introduction

Infection is a common and serious complication of cancer treatment in children that often presents as fever with neutropenia, a condition termed febrile neutropenia (FN). Because prolonged neutropenia is a risk factor for severe infections, the recommendations for children with that condition is to immediately initiate treatment with empirical broad-spectrum antibiotics [1]. Despite that, bacterial bloodstream infections are detected in only 10–30% of FN episodes, the majority of which are of unknown etiology [2,3,4]. Previous studies detected common respiratory viruses in up to 40–50% of children with FN [3,5,6]; however, the presence of a virus does not always correspond to an acute infection, as respiratory viruses can be detected in asymptomatic children [7]. In addition, some children may suffer from fever induced by their tumor or chemotherapy (drug), which further complicates the interpretation of the etiology of FN. As a consequence, it is challenging to determine when to use antibiotic treatment in children with FN, which typically involves administration of broad-spectrum antibiotics for 5 to 10 days [2,8]. Now when antibiotic resistance is a raising global health problem, there is an urgent need to find better diagnostic tools that can reduce the overuse of broad-spectrum antibiotics.

Infectious microbes have a number of signature molecules, referred to as pathogen-associated molecular patterns (PAMPs), that are essential for their survival and pathogenicity. The immune system recognizes PAMPs through highly conserved pattern-recognition receptors (PRRs), that are predominantly expressed by innate immune cells such as the epithelial lining, monocytes, dendritic cells and neutrophils. When PAMPs are recognized by the immune system, innate immune cells rapidly launch anti-microbial responses tailored to eliminate different categories of pathogens by producing inflammatory cytokines, chemokines and type I interferons (IFNs). Ultimately, those innate immune responses trigger and shape long-lasting adaptive immunity [9].

Gene-expression profiling techniques have initiated a new era in the identification of prognostic and diagnostic biomarkers of infectious disease. Aside from pathogen-related factors, host responses to infectious agents can be examined on a genome-wide scale to reveal transcriptional signatures that associate to innate-response signaling pathways [10]. By assessing immune-related gene-expression profiles, Hu et al. were able to distinguish children with asymptomatic and symptomatic viral infection, as well as those with bacterial infection [11]. In addition, the study also identified that children with asymptomatic viral infection were indistinguishable from their afebrile uninfected peers. A subsequent study identified a 2-script host-RNA signature that could differentiate between viral and bacterial infections with high sensitivity and specificity rate [12]. However, even if the new techniques are promising, they remain on an experimental level; to be implemented as a diagnostic tool in the clinic, further studies on different patient cohorts are necessary.

In previous studies of diagnostic gene-expression profiling in febrile children, all participants were immunocompetent [11,12]. Hence, little is known about gene-expression profiles in children whose immune response is altered by cancer treatment. Therefore, we asked whether diagnostic gene-expression profiling is feasible in children with FN who are undergoing cancer treatment. We examined the blood transcriptomes of a cohort of children that presented with FN during cancer treatment, which was clinically attributed to viral, bacterial, co-infection or unknown etiology.

## 2. Results

A total of 63 FN episodes from 45 patients were included in the final analysis (Figure 1). The RNA samples collected in 43 of the FN episodes (68%) were sufficient for RNA-sequencing (RNA-seq). The characteristics of the episodes with (*n* = 43) and without (*n* = 20) sufficient RNA for sequencing are outlined in Table 1. The episodes with insufficient RNA for sequencing had lower white blood cell counts (WBCs) and absolute neutrophil counts (ANCs) on average than those with sufficient RNA for sequencing (*p* < 0.001 for each comparison). Samples from children with hematological malignancies tended to contain insufficient RNA for sequencing (*p* = 0.06). Furthermore, episodes with insufficient RNA for sequencing had higher C-reactive protein (CRP) levels (*p <* 0.01), and more days with broad-spectrum antibiotic treatment (*p* ≤ 0.05) compared with episodes with sufficient RNA for sequencing.

### 2.1. Clinical and Microbiological Findings in the Episodes with Sufficient RNA for Sequencing

The 43 episodes with sufficient RNA for sequencing originated from 35 individuals. The etiology was classified as respiratory virus infection in 15 episodes, unknown in 22 episodes, co-infection in 4 episodes, and bacterial infection in 2 episodes (Figure 1). Twelve control samples were obtained from 12 different individuals. Nine of those were control samples obtained after an FN episode with sufficient RNA for sequencing (four episodes with viral etiology and five episodes with unknown etiology). The remaining three samples were control samples obtained after an FN episode with insufficient RNA for sequencing. The characteristics of the study and control samples are outlined in Table 2. The control samples had higher WBCs and ANCs than the samples with viral infection (*p* < 0.001 and *p* < 0.001, respectively) or unknown etiology, (*p* ≤ 0.05, *p* < 0.001, respectively), and higher ANCs, but not WBCs than the samples with co-infection (*p* < 0.01). No other differences were seen among the cohorts. Respiratory symptoms were present in 87% and 100% of the episodes with viral or co-infection etiology, respectively, but were less common in the episodes with unknown etiology or bacterial etiology (64% and 50%, respectively) (Table 2). Gastrointestinal-symptoms and local symptom were uncommon in all groups.

### 2.2. Pathogen-Related Blood Transcriptomes in Immunosuppressed Children with Neutropenia

To determine whether it is possible to detect pathogen-specific immune responses on the basis of gene expression in immunosuppressed children, we compared the blood transcriptomes of samples with viral infection, bacterial infection, co-infection, or unknown etiology with those of the control samples. The differentially expressed genes identified in the viral-, co-infection and unknown etiology are shown in Figure 2A. The up- and down-regulated genes for the comparisons, and similarities and differences amongst them, are visualized in Figure 2B. In the samples with viral infection and unknown etiology, the majority of the differentially expressed genes were upregulated; 178 out of 231 and 261 out of 442, respectively. In the samples with co-infection, the majority of the deviating genes expressed was down-regulated; 124 out of 219. When comparing samples with bacterial infection with the control group, only two genes were differentially expressed (data not shown).

Next, we performed pathway analysis of the differentially expressed genes identified in the viral, unknown, and co-infection etiologies. The top five ranked canonical pathways that were up- and down-regulated respectively, in each etiology are outlined in Figure 2C. The up-regulated genes in the three etiologies were enriched with cellular processes related to proliferation and cellular stress response, with no clear enrichment with innate responses to pathogens (Figure 2C). Among the down-regulated genes in the three etiologies, few genes clustering into pathways connected to responses to infection, such as IFN in the co-infection etiology and helper T cell activation signaling in samples with viral infection (Figure 2C). To investigate if there were any group-specific pathways that were activated, a new pathway analysis was performed on group-specific genes (viral, *n* = 75, unknown *n* = 265, and co-infection *n* = 149, Figure 2A) and only few genes clustered into any pathway in samples with viral infection etiology, while the resulting top canonical pathways for samples with unknown and co-infection etiology were identical (data not shown).

Gene expression was further compared, in a pair-wise manner, among the viral, unknown and co-infection etiologies. The pair-wise comparisons did, however, not reveal any differentially expressed genes. On the other hand, there were 47, 7 and 32 genes that were differentially expressed between the bacterial etiology and the viral, unknown, and co-infection etiologies, respectively, which included the Sialic acid-binding immunoglobulin-type lectin (siglec) family of molecules (Appendix A).

## 3. Discussion

In this study, we asked whether it is possible to use the pathogen specific innate transcriptome as a diagnostic tool in immunosuppressed children with FN. For this purpose, we examined a relatively large sample set from clinically well-characterized children that presented with FN during the course of cancer treatment. Despite the relatively low RNA-input requirements, the host transcriptome did not reliably distinguish different types of pathogens in children with FN.

To explore gene-expression profiles, it is critical to obtain sufficient amounts of RNA for sequencing. Insufficient RNA has not been a problem in earlier studies on non-immune suppressed patients that used RNA sequencing for gene expression profiling [13]. In an earlier study on patients presenting with FN EdgeSeq was used instead of RNAseq due to the lack of WBC [14]. In the present study, there was insufficient RNA for sequencing in 33% of the episodes, likely due to lower WBCs and ANCs in the samples with insufficient RNA compared with those in the samples with sufficient RNA. That finding is of great importance as it implies that in children with the highest risk of severe bacterial infections [15], due to low levels of circulating leukocytes, gene expression profiling is not feasible. The risk of severe infection is high in patients with an ANC of 0.5 × 10^9^/L and increases with ANC < 0.1 × 10^9^/L [15,16]. In the samples with insufficient RNA for sequencing in our study, the median WBC was 0.3 × 10^9^/L. Furthermore, episodes with insufficient RNA for sequencing and ANC < 0.1 × 10^9^/L had more days on antibiotics, higher CRP, and a trend towards more bacterial infections (15% vs. 5%) compared with the episodes with sufficient RNA for sequencing. This clearly demonstrate that host transcriptional profiles are not suitable for the detection of specific microbial infections in high-risk patients with FN.

We and other researchers have identified respiratory viruses as a common agent in episodes of FN with rhinovirus, influenza virus and respiratory syncytial virus being the most common detected viruses [3,17,18]. It is still unclear, however, if the detected viruses are the causative agents of FN, because PCR techniques can detect viruses from earlier infections, and viruses can also be detected in asymptomatic children [7]. An alternative way to assess etiology might be to examine the host immune response to specific pathogens. In that regard, studies have shown that the immune responses in children with asymptomatic viral infections differ from those children with symptomatic viral infections that display activated innate signaling [11,19]. In our cohort, no pathogen-specific canonical pathways were activated in the samples with virus infection in comparison with the control samples. The lack of a pathogen-specific response in the samples from patients with viral infections might reflect viral DNA/RNA from a previous infection that was unrelated to the FN episode. In all patients, there was a possibility that the cause of the fever was tumor and/or drug related. The majority of the patients presented with focal symptoms, that were mainly associated with the respiratory tract, indicating an active infection. Therefore, the most likely explanation for the lack of pathogen-specific pathway activation is the low numbers of immune cells in the blood. Even in such cases, the local mucosal innate response generated by the infected cells might still be intact. This has indeed been shown in a study comparing host gene expression in blood and nose [20]. Therefore, to further address the etiological role of respiratory virus in children with FN, future studies should include gene-expression analyses of secretions representing relevant mucosal sites.

When comparing samples with unknown etiology with the control samples, we saw the same pattern of no apparent immune response in the blood. By contrast, in the samples with co-infection (respiratory virus and bacteria), we observed a larger set of downregulated genes and a down-regulated innate immune response. These episodes differed from the unknown and viral episodes as they occurred in children with a high WBC count, despite also having a low ANC count. Therefore, the gene expression signals are probably generated from other subgroups of immune cells such as lymphocytes and could theoretically be one reason for the differences seen in this group as compared to the others. Downregulation of immune cell responses in children suffering from co-infection infections may reflect migration and redistribution of effector cells to the affected tissue, and thus, a corresponding lack of them in the circulation. In addition, neutrophils are key players in the early innate response to invading pathogens and low ANC could thus facilitate host colonization by multiple pathogens. Whether the dampened gene expression profile in the patients with co-infection is related to cancer-related or treatment-related immunosuppression, redistribution of cells from the circulation to peripheral tissue or to pathogen modulation of the innate immune response remains an open question.

When comparing the group with bacteria with the control group, only two genes were differently expressed. This could be due to the small sample size and the rather stringent statistical methods used. The high infection parameters support the bacterial etiology.

Earlier studies of immunocompetent children with microbiological documented infections identified pathogen-specific host transcriptional profiles that could differentiate between viral and bacterial infections, which has since been called a paradigm shift in how to address etiology in patients with an infection [11,12,13,21]. However, before implementation as diagnostic tools in the clinic, accessible and time effective diagnostic platforms needs to be developed and validated on a large-scale [22]. In addition, even if the findings so far have been reproducible in children of different genetic backgrounds and for various pathogens [21,23], these findings still need to be investigated in groups of children with potential impaired immune system at risk for severe infections. Thus, further studies should also focus on children treated with immune suppressant drugs, children in low income countries with malnutrition and children in areas with a high incidence of chronic infectious diseases, such as HIV or tuberculosis [22,24]. In the present study, there were no differences in gene expression detected among the samples with virus infection, co-infection, and unknown etiology. A set of genes correlated with the neutrophil response to bacteria was differentially expressed, between those samples and the samples with bacterial infection, although the latter consisted of only two samples. To our knowledge, only one previous study has investigated the gene expression signals in adult patients presenting with FN. Kelly et al. investigated the gene expression profile of 2560 genes related to cancer and immunology using EdgeSeq [14]. They compared the transcriptome profiles of seven patients with FN to 22 controls with fever of unknown origin and, in addition, metabolomic profiling from the same cohort was performed. They could identify a set of three genes that correlated to a bacterial response; however, it is difficult to make any comparisons to the present study as different methods were applied [14]. Importantly, our data also suggest that the heterogeneity of cells (neutropenia) impact gene expression signals. Hu et al. investigated the association of gene expression level with total WBC and observed no correlation [11]. However, they did find correlations between the expression levels of some genes and the counts of sub-groups of white blood cells, such as monocytes, neutrophils and lymphocytes [11]. In our study, the WBCs and ANCs were low in all of patients because of immunosuppression, and differential counts of monocytes and lymphocytes were not taken because of local routines. In addition, the function of the circulating immune cells might be affected by chemotherapy, making it difficult to draw definite conclusions on gene expression.

We acknowledge the following limitations to our study: First, our cohort was heterogeneous in terms of the characteristics of the underlying cancers and the different treatment regimens used, both of which might impact the host immune responses. Second, transcriptional profiles investigating infectious etiology is a new technique and has not been used on a large scale before in immune suppressed children. In one-third of the episodes had RNA-concentrations that were too low for RNA-sequencing and were mirrored by low WBCs and/or ANCs. Those factors are correlated with an increased risk of bacterial infections that imposed a selection bias against inclusion of cases with bacterial infections. To our knowledge, however, our study is the first to assess gene expression signatures against pathogens in a large number of pediatric oncology cases on immunosuppressive treatment, which are clinically well-defined, to understand if there is a connection between the blood transcriptome and pathogen type in FN.

## 4. Materials and Methods

### 4.1. Study Cohort

Patients were enrolled at the Childhood Cancer unit at Astrid Lindgren Children’s Hospital in Stockholm between January 2013 and June 2014 [8]. All care takers to children (0–18 years of age), who were undergoing chemotherapy and met the criteria for FN were asked to participate. FN was defined as a body temperature ≥38.5 °C on one occasion or ≥38.0 °C on two occasions at least 60 min apart, combined with an absolute neutrophil count of either ≤0.5 × 10^9^/L on one occasion or ≤1.0 × 10^9^/L with a decline to less than ≤0.5 × 10^9^/L over a subsequent 48 h period. The same individual could be enrolled multiple times if he or she experienced recurrent episodes of FN during the study period, with the prerequisite that the child had been afebrile for more than 72 h and had completed his or her antibiotic treatment for the previous episode of FN. Children included within 72 h of fever onset and from whom a complete set of samples were collected (nasopharyngeal aspirate (NPA), blood culture, and PAXgene tube) were eligible. After discharge from the hospital, each child was asked to leave a convalescence sample at the next scheduled appointment at the hospital, further addressed in the study as control samples. No time limit was set for when the control sample was collected, but it was defined as a control sample only if the child lacked any clinical sign of infection at the time the sample was taken. In addition, in the cases where a respiratory virus was detected, the virus must have been cleared (negative NPA) at the time of control sampling. Clinical data, including age, gender, type of malignancy, treatment intensity, symptoms of infection, days with antibiotics and days with fever and the duration of hospitalization, were collected from the medical records. In total, blood samples taken during 67 FN episodes met the study criteria and were included in the study. Further, four of those samples were excluded because RNA was isolated more than 72 h after the onset of fever and the final study cohort consisted of 63 FN episodes representing 45 individuals (Figure 1). Oral and written information regarding the study were provided to caretakers prior to enrollment, and signed consent was obtained from the participants’ caretakers. In the cases, the children had sufficient reading and comprehension abilities the children were provided with a simplified version of study information. The study was approved by the Regional Ethical Review Board in Stockholm; 2008/648-31/4 and 2009/286-32.

### 4.2. Blood Sampling, Microbiological Testing and RNA Preparation

Blood samples were collected and stored at −20 °C in a PAXgene Blood RNA Tube (PreAnalytiX, Homebrechtikon, Switzerland). White blood counts (WBC) and absolute neutrophil counts (ANC) were analyzed at the Karolinska University Laboratory. For the definition of the study criteria FN the ANC counts at time of admission was used. WBC/ANC counts used as comparison between the study groups was sampled the same day as the RNA sample. In addition, NPA was collected and directly analyzed with in-house real-time PCRs for the following 16 viruses: adenovirus; bocavirus; coronaviruses NL63/OC43/229E/HKU1; enterovirus; influenza virus A, including A(H1N1)pdm09 and B; metapneumovirus; parainfluenza viruses 1-3; respiratory syncytial virus; and rhinovirus [25]. As per clinical routine, blood cultures (1–10 mL) were collected from all children and analyzed at the Karolinska University Laboratory. Control blood samples were collected as described above, and when a child tested positive for a respiratory virus at inclusion, a follow-up NPA was collected and analyzed in the same manner as the first sample [25]. RNA was extracted using the PAXgene® Blood RNA system kit (PreAnalytiX) according to manufacturer’s instruction. The concentration and purity of the extracted RNA [RNA integrity number (RIN)] were measured using a NanoDrop ND-1000 spectrophotometer (Thermo Fisher Scientific, MA, USA) and an RNA ScreenTape Assay (on an Agilent 2200 Tapestation, Agilent Technologies, Santa Clara, CA, USA), respectively, according to the manufacturers’ instructions. Generally, an RIN > 7 indicated good sample quality.

### 4.3. Microbiological Documented Infections and Grouping of Patients

Patients with confirmed virus or bacterial infections at the time of their FN episode were considered to have a microbiologically documented infection (MDI). Patients with a viral MDI were those in whom a respiratory virus was detected by virus-specific PCR. Testing for other viruses, such as gastro-intestinal viruses and herpes virus, was guided by clinical symptoms. Positive findings for non-respiratory viruses were included as MDIs if the treating clinician considered them clinically relevant to the FN episode. Bacterial infections were diagnosed on the basis of a positive blood culture. Positive blood cultures determined to be either contaminants, or not clinically relevant, by the treating clinician and/or the laboratory were not considered positive findings of MDI. In addition, bacteria cultured from local foci that appeared with other FN symptoms and/or were determined to be of clinical relevance by the treating clinician were considered bacterial MDIs. Co-infection was defined as coincident viral and bacterial MDIs; this was confirmed using the same criteria as described above. Patients with unknown FN etiology had no MDI, according to the criteria described above.

### 4.4. RNA Preparation, Sequencing and Analysis

Sequencing libraries were constructed using an Illumina TruSeq Stranded kit (San Diego, CA, USA). A specific sample preparation protocol was performed that included mRNA isolation, RNA fragmentation, cDNA synthesis, ligation of adapters (barcode and binding spots) and cDNA amplification according to the manufacturer’s instructions. The index cDNA was normalized, samples were combined into a pool, and the pool (containing all specifically marked samples) was sequenced using an Illumina Nextseq 550 with 75-cycle v2 sequencing, generating 75-bp single-end reads (San Diego, CA, USA). The reads were mapped to a human reference genome, GRch38 and the mapping data were normalized using DESeq2. A false discovery rate (FDR) *<* 0.05 and a fold change (FC) difference of at least 1.5 were used to identify genes that were differentially expressed between groups with different MDI status and controls. SeqMonk (version 1.46.0) was used to perform the data analysis. Pathway analysis was assessed by Ingenuity Pathway Analysis software (https://digitalinsights.qiagen.com/products-overview/discovery-insights-portfolio/analysis-and-visualization/qiagen-ipa/; content version: 51963813; release date: 2020-03-11; Ingenuity Systems, Redwood City, CA, USA). Circos software package (http://circos.ca/) was used to visualize the genomic data in a circular plot [26]. Raw data can be assessed at the GEO data base, accession number; GSE152341.

Fisher’s exact tests and the Mann–Whitney U tests were used for group comparisons of categorical and continuous data, respectively. A *p*-value ≤ 0.05 was considered statistically significant. Data were analyzed using GraphPad Prism (GraphPad Prism, San Diego, CA, USA). Blood count levels reported as “<0.1” from the laboratory was uniformly set to 0.01 when calculating statistics.

## 5. Conclusions

Differences in pathogen-specific host transcriptional profiles between bacterial infections and viral infections are suggested as diagnostic tools in the clinic. Our results suggest that they are not suitable for determining the etiology of FN in immunosuppressed children during cancer treatment, because children with low WBCs or ANCs and, hence, an elevated risk of infection, have too few immune cells in their blood for reliable gene expression analysis. Therefore, future studies should investigate the relationship between circulating counts of immune cells and gene expression levels, both in blood and at the local infection site before diagnostic gene expression profiling is implemented in the clinic.

## Figures and Tables

**Figure 1 ijms-21-05305-f001:**
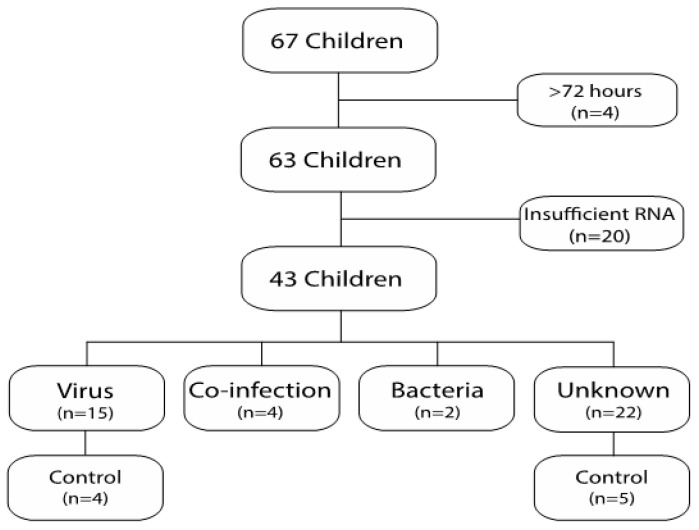
Flowchart of included patients.

**Figure 2 ijms-21-05305-f002:**
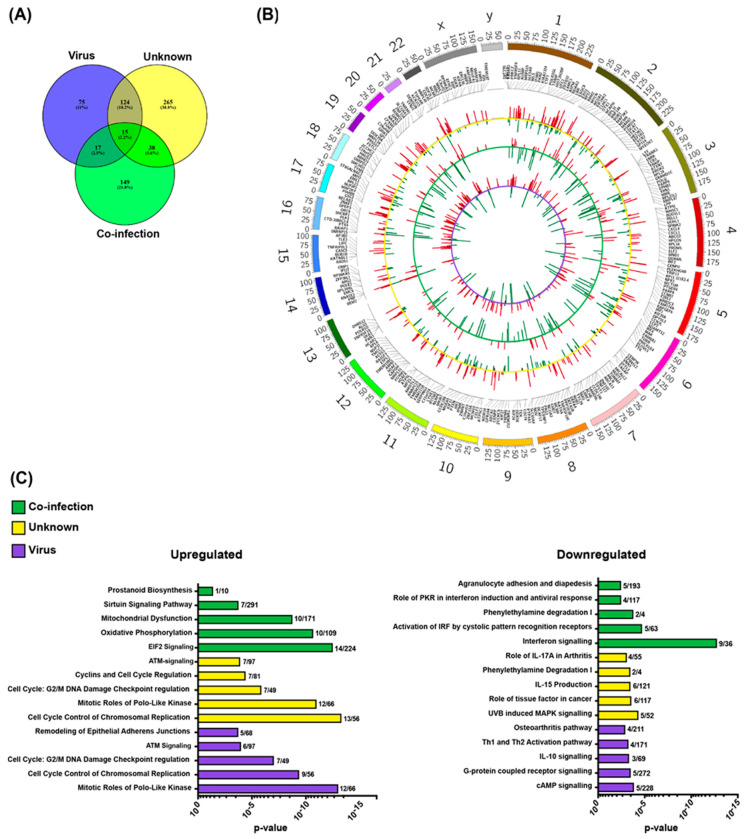
Differentially expressed genes in the co-infection, unknown and virus group during febrile neutropenia as compared to the control samples. (**A**) Number of overlapping and non-overlapping transcribed genes that were markedly different between the groups. (**B**) Circos plot depicting up- and down-regulated genes for the three comparisons (virus vs. control = inner violet circle; co-infection vs control = middle green circle and unknown vs control = outer yellow circle). (**C**) The columns represent the top five up- (left) and down-regulated (right) significant canonical pathways for each group after Ingenuity Pathway Analysis. The numbers after the columns represent number of genes up- and down-regulated in each network.

**Table 1 ijms-21-05305-t001:** Episodes with and without sufficient RNA for sequencing (RNA-seq).

	Sufficient RNA for RNA-seq *N* = 43	Insufficient RNA for RNA-seq *N* = 20	*p*-Value
Age, median (range)	7.6, (0.8–16.1)	8.25, (0.9–12.3)	0.83
Gender (*n* girl) (%)	23 (53)	10 (50)	1
Hematological malignancy, *n* (%)	16 (37)	13 (65)	0.06
WBC ^a^ (10^9^/L), median (range)	**1.6 (0.2–14.6)**	**0.3 (0.1–0.7)**	**<0.001**
ANC ^b^ (10^9^/L), median (range)	**0.1 (<0.1 ^c^–10.9)**	**<0.1 ^c^ (<0.1–<0.1)**	**<0.001**
Temp max, median (range)	39.2 (38.1–40.5)	39.0 (38.1–40.2)	0.67
CRP max, median (range)	**73 (4–412)**	**117 (41–341)**	**<0.01**
Days with antibiotics, median (range)	**7 (0–30)**	**8.5 (5–17)**	**<0.05**
Respiratory virus infection, *n* (%)	15 (35)	3 (15)	0.14
Unknown etiology, *n* (%)	22 (51)	13 (65)	0.42
Co-infection, *n* (%)	4 (9)	1 (5)	1.0
Bacterial infection, *n* (%)	2 (5)	3 (15)	0.32

Statistically significant differences are highlighted in bold. ^a^ Hematological malignances included; acute lymphoblastic leukemia, acute myeloid leukemia, and non-Hodgkin lymphoma, ^b^ WBC and ANC counts was collected the same day as the RNA samples. ^c^ ANC below 0.1 was reported from the laboratory as “<0.1”. Abbreviations: WBC: white blood cells; ANC: absolute neutrophil counts; CRP: C-reactive protein.

**Table 2 ijms-21-05305-t002:** Clinical characteristics of the RNA-sequenced study and control cohorts.

	Viral Infection *^,a^ *n* = 15	Unknown Etiology *^,b^ *n* = 22	Co-Infection * *n* = 4	Bacterial Infection *^,c^ *n* = 2	Control * *n* = 12
Age (Median, range; IQR)	9.9 (0.8–16.1)	7.5 (0.5–16.0)	7.1 (3.4–10.1)	3.0 (1.5–4.5)	9.7 (0.6–15.8)
Gender (*n* girl) (%)	10 (67)	11 (50)	1 (25)	1 (50)	8 (67)
Hematological ^d^, *n* (%)	6 (40)	8 (36)	0	2 (100)	5 (42)
Hight intensity treatment *n* (%)	6 (40)	9 (41)	2 (50)	2 (100)	6 (50)
Medium intensity treatment, *n* (%)	9 (60)	13 (59)	2 (50)	0	6 (50)
WBC ^e^ count (10^9^/L) (Median, range)	**1.3 (0.2–3.8)**	**1.7 (0.2–14.6)**	3.1 (1.3–4.5)	1.0 (0.3–1.6)	**3.7 (0.7–0.6)**
ANC ^e,f^ (10^9^/L) (Median, range)	**0.1 (<0.1–1.3)**	**0.2 (<0.1–10.9)**	**<0.1 (<0.1–0.4)**	<0.1 (<0.1–<0.1)	**2.1 (0.1–5.6)**
Temp max (median, range)	39.1 (38.1–40.5)	39.1 (38.1–40.2)	39.2 (38.6–39.8)	39.2 (39.4–40.3)	N/A
CRP max (median, range)	66 (6–240)	67 (4–412)	86 (22–257)	174 (141–207)	N/A
Days with neutropenia ^g^ (median range)	6 (2–37)	9 (1–>30)	5.5 (4–13)	255 (20–>30)	N/A
Days with fever (median, range)	2 (1–6)	2 (1–16)	1.5 (1–3)	3 (2–4)	N/A
Days with antibiotics (median range)	7 (0–10)	7 (0–30)	8 (7–13)	15.5 (14–17)	N/A
Days at hospital (median, range)	5 (0–9)	5 (2–30)	5 (3–17)	9.5 (8–11)	N/A
Respiratory symptoms, n (%)	13 (87)	14 (64)	4 (100)	1 (50)	N/A
Gastrointestinal symptoms, *n* (%)	3 (20)	3 (14)	2 (50)	0 (0)	N/A
Local symptoms, *n* (%)	1 (7)	2 (9)	2 (50)	1 (50)	N/A

* Statistical analyses of continuous data were only calculated among the viral-, unknown-, co-infection- and control groups. Statistical analyses of categorical data were calculated among the viral, unknown and control groups. Statistical differences are highlighted in bold. ^a^ Viral infections consisted of 8 rhinovirus, 3 coronavirus, 2 Flu A, 1 Flu B, 1 respiratory syncytial virus and 1 parainfluenza 3. In one episode, two viruses were detected (rhino and corona). ^b^ Co-infections consisted of parainfluenza virus detected with clostridium toxin B (feces), rhinovirus detected with pseudomonas aeruginosa (wound), coronavirus detected with staphylococcus aureus (wound), and bocavirus detected with alpha streptococcus (blood). ^c^ Staphylococcus epidermis and Escherichia coli were each detected separately in blood samples. ^d^ Hematological malignances included; acute lymphoblastic leukemia, acute myeloid leukemia and non-Hodgkin lymphoma. ^e^ WBC and ANC counts were collected the same day as the RNA samples ^f^ ANC below.0,1 was reported from the laboratory as “<0.1”. ^g^ In two patients the exact number of days with neutropenia was not possibly to calculate because of too infrequent sampling. They are stated as > 30 days. Abbreviations: WBC, white blood cells; ANC, absolute neutrophil counts; CRP, C-reactive protein.

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
