# Peer review of "The Feasibility of Host Transcriptome Profiling as a Diagnostic Tool for Microbial Etiology in Childhood Cancer Patients with Febrile Neutropenia"

_ijms, 2020, doi:10.3390/ijms21155305_

Round 1

Reviewer 1 Report

I read this manuscript with interest and I consider it well written.

I appreciate a lot the idea and research question. I suggest to publish it after minor suggestions:

  1. Introduction: very well written. 
  2. Methods: I believe there is a mistake in text formatting. Please add methods section before results. For this part of manuscript I do not have suggestions. 
  3. Results: no comments
  4. Discussion: consider to cite this article (Marotta C et al. The At Risk Child Clinic (ARCC): 3 Years of Health Activities in Support of the Most Vulnerable Children in Beira, Mozambique. Int J Environ Res Public Health. 2018;15(7):1350. Published 2018 Jun 27. doi:10.3390/ijerph15071350) to explain how the most vulnerable patients are defined "children at risk" and why it needs more clinical but also research attention.

Author Response

I read this manuscript with interest and I consider it well written.

I appreciate the idea and a research question a lot. I suggest to publish it after minor suggestions:

  1. Introduction: very well written. 
  2. Methods: I believe there is a mistake in text formatting. Please add methods section before the results. For this part of manuscript I do not have more suggestions. 
  3. Results: no comments
  4. Discussion: consider to cite this article (Marotta C et al. The At Risk Child Clinic (ARCC): 3 Years of Health Activities in Support of the Most Vulnerable Children in Beira, Mozambique. Int J Environ Res Public Health. 2018;15(7):1350. Published 2018 Jun 27. doi:10.3390/ijerph15071350) to explain how the most vulnerable patients are defined "children at risk" and why it needs more clinical but also research attention.

Answer; Thank you for reviewing our paper.

  1. Formatting and method section are written according to the journals guidelines.
  1. We have now added the above mentioned paper and discussed it, please see line 230-232.

Reviewer 2 Report

The authors write an interesting study about a current ID topic: the possibility of using host transcriptome profiling to discriminate between the type of infection ; this article in particular is focused on the specific application of this technique in patients with fever and neutropenia.

1) While host transcriptome profiling is surely a promising strategy, to my knowledge it remains completely experimental at this stage, with no clinical application so far. This should be put in the introduction and in the limitations.

2) following what is written above, on lines 211-213: these sentences should be tuned down.

3) there are important papers in this field that were not mentioned, like: 

Holcomb et al, Host based peripheral blood gene expression analysis for the diagnosis of infectious disease, journal of clinical Microbiology, 2016, Kelly et al: Integrative omics to detect bacteremia in patients with FN. PLOS one 2018

4) The authors should interpret more their findings; it would be useful to have some hypothesis to explain why, for example, " when comparing samples with bacterial infection with the control group, only 2 genes were differentially expressed" ( line 130), whereas " in the samples with co-infection, the majority of the deviating genes expressed was down-regulated" ( line 129). 

Author Response

The authors wrote an interesting study about a current ID topic: the possibility of using host transcriptome profiling to discriminate between the types of infections. This article in particular is focused on the specific application of this technique in patients with fever and neutropenia.

1)While host transcriptome profiling is surely a promising strategy, to my knowledge it remains completely experimental at this stage, with no clinical application so far. This should be put in the introduction and in the limitations.

2) following what is written above, on lines 211-213: these sentences should be tuned down.

3) there are important papers in this field that were not mentioned, like: 

Holcomb et al, Host based peripheral blood gene expression analysis for the diagnosis of infectious disease, journal of clinical Microbiology, 2016, Kelly et al: Integrative omics to detect bacteremia in patients with FN. PLOS one 2018

4) The authors should further interpret their findings: it would be useful to have some hypothesis to explain why, for example "when comparing samples with bacterial infection with the control group, only 2 genes were differentially expressed" ( line 130), whereas " in the samples with co-infection, the majority of the deviating genes expressed was down-regulated" ( line 129). 

 Answer; Thank you for reviewing the paper and the valuable comments that helped to improve the paper.

  1. This is now added in the introduction and limitation.
  2. This paragraph is now –re-written and we added the “Holcomb et al” reference.
  3. Thanks for pointing out these papers. Now added.
  4. It is now discussed, please see line 237-241 and 208-211 in the discussion part.

Reviewer 3 Report

The authors aimed to examine whether gene expression profiling  was feasible in a study sample of children with febrile neutropenia ( FN) who were undergoing cancer treatment.

The methodology is well described with enough experimental data and results to support the work.

The study is easy to follow and covers an hot topic, but some issues should be improved before publication. Several typos should be corrected through the text.

Please refer to statics with p-value in italics.

Please provide the number of the Regional Ethical Review Board in Stockholm.

Discussion Section will be very useful for the readers to add a sentence about the promising results of  oral probiotics and a potential tool that influence infection as oral and respiratory tract infection in pediatric population and the related nutrition absorption capacities( please see and discuss PMID:30536353; PMID:31646599)

Author Response

The authors aimed to examine whether gene expression profiling  was feasible in a study sample of children with febrile neutropenia ( FN) who were undergoing cancer treatment.
The methodology is well described with enough experimental data and results to support the work.
The study is easy to follow and covers an hot topic, but some issues should be improved before publication. Several typos should be corrected through the text.

Please refer to statics with p-value in italics.

Please provide the number of the Regional Ethical Review Board in Stockholm.

Discussion Section will be very useful for the readers to add a sentence about the promising results of  oral probiotics and a potential tool that influence infection as oral and respiratory tract infection in pediatric population and the related nutrition absorption capacities( please see and discuss PMID:30536353; PMID:31646599)

Answer: Thank you for reviewing our paper.

  1. “but some issues should be improved before publication” Please specify if there is anything more than the points below that should be addressed. We have read the paper and corrected the typos we found.
  2. Ethical number is now added and p-values in italics.
  3. As for the suggested papers to refer to we believe these are very interesting but out of the scoop of the paper. In addition, our cohort is a group of immune suppressed children. The usage of probiotics in this cohort should be further investigated.

Round 2

Reviewer 2 Report

Thanks for having addressed my comments . 

Reviewer 3 Report

The methodology is well described with enough experimental and interesting results to support the work. I appreciate that the required changes were made in a little days.